# Pan-cancer proteogenomic investigations identify post-transcriptional kinase targets

Abdulkadir Elmas [1], Serena Tharakan[1], Suraj Jaladanki [1], Matthew D. Galsky[2], Tao Liu [3] & Kuan-lin Huang [1✉]

Identifying genomic alterations of cancer proteins has guided the development of targeted therapies, but proteomic analyses are required to validate and reveal new treatment opportunities. Herein, we develop a new algorithm, OPPTI, to discover overexpressed kinase proteins across 10 cancer types using global mass spectrometry proteomics data of 1,071 cases. OPPTI outperforms existing methods by leveraging multiple co-expressed markers to identify targets overexpressed in a subset of tumors. OPPTI-identified overexpression of ERBB2 and EGFR proteins correlates with genomic amplifications, while CDK4/6, PDK1, and MET protein overexpression frequently occur without corresponding DNA- and RNA-level alterations. Analyzing CRISPR screen data, we confirm expression-driven dependencies of multiple currently-druggable and new target kinases whose expressions are validated by immunochemistry. Identified kinases are further associated with up-regulated phosphorylation levels of corresponding signaling pathways. Collectively, our results reveal protein-level aberrations—sometimes not observed by genomics—represent cancer vulnerabilities that may be targeted in precision oncology.

[1] Center for Transformative Disease Modeling, Department of Genetics and Genomic Sciences, Tisch Cancer Institute, Icahn Institute for Data Science and Genomic Technology, Icahn School of Medicine at Mount Sinai, New York, NY 10029, USA. [2] Division of Hematology and Medical Oncology, Tisch Cancer Institute, Icahn School of Medicine at Mount Sinai, New York, NY 10029, USA. [3] Biological Sciences Division, Pacific Northwest National Laboratory, Richland, WA 99354, USA. ✉email: kuan-lin.huang@mssm.edu

Personalized medicine provides tailored treatment plans based on each tumor's unique genomic or protein bio-markers. Large-scale patient–cohort studies across cancer types have identified driver genomic alterations that can be targeted, including amplification, mutations, and fusions, particularly enriched in oncogenic kinases[1]. However, observations of genomic alterations often presume downstream molecular effects that may not be validated in patient samples[2]. Further, protein- and phosphorylation-level aberrations can arise posttranscriptionally or posttranslationally[3,4]. Lastly, druggable mutations are not found in substantial fractions of tumors in multiple cancer types[5,6]. Comprehensive proteomic analyses are required to validate genomic findings and discover new kinase protein targets.

Recent advancements in mass spectrometry (MS) have enabled the characterization of the majority of expressed proteomes. Efforts including the Clinical Proteomic Tumor Analysis Consortium (CPTAC)[2] have generated an expanding collection of global proteomic cohorts that quantified over 10K proteins and 30 phosphorylation sites (phosphosites) in multiple cancer types. These rich proteomic data sets provide ample opportunities to identify aberrant proteins as treatment targets and compare findings across cancer types, but most analyses have remained limited to single cancer types.

Herein, we curated global MS proteomics and phosphoproteomics data from ten recently characterized cancer cohorts, conducting a pan-cancer study totaling 1071 cases. We identified diverse activation patterns in oncogenic signaling pathways across cancer types. Systematic analyses of overexpressed kinase proteins identified 23 overexpressed, druggable pan-cancer or cancer-specific targets, several of which being not readily observed by genomics at the DNA or mRNA level. By overlaying CRISPR vulnerability screen analyses of cancer cells in corresponding lineages, we discovered overexpressed protein kinase targets that show genetic dependencies. Our results highlight the importance of proteome-based identification of targeted treatment options for patients, particularly in those without actionable mutations.

## Results

**Assembly of a pan-cancer global proteomics cohort**. The pan-cancer proteomic study combined a multitude of analyses to identify and validate actionable kinase proteins found in high-throughput proteomic cohorts (Fig. 1a). We first curated recently characterized global MS proteomics data sets of 1071 cases across 10 cancer types. These include seven projects affiliated with the National Cancer Institute's CPTAC (NCI CPTAC): 115 cases of breast cancer (BRCA), 84 cases of ovarian cancer (OV), 97 cases of uterine corpus endometrial carcinoma (UCEC)[7], 95 cases of colorectal cancer (CRC)[8], 110 cases of clear cell renal cell carcinoma (CCRCC)[9], 80 cases of early-onset gastric cancer (stomach adenocarcinoma, STAD)[10], and 109 cases of lung adenocarcinoma (LUAD). The other studies included 101 cases of hepatitis B virus-related hepatocellular carcinoma (HCC)[11], 76 cases of localized prostate cancer (PRAD)[12], and 45 medulloblastoma cases (MB)[13]. All studies used isotope labeling[14] and thus quantified the relative abundance of protein/phosphosites. We compiled available clinical information from each of these studies, including gender, age, and clinical stage (Fig. 1c).

We applied standardized normalization and quality-control criteria to each of the individual data sets ("Methods"). Based on a previously curated list of 683 kinase proteins[15,16], these data sets include an average of 437 detected kinase proteins (Fig. 1b). Six cancer cohorts, namely BRCA, OV, UCEC, CRC, CCRCC, and LUAD, also included phosphoproteomics data (Fig. 1c). We further identified 409 currently druggable kinases in this data set

by overlapping the list of kinase proteins with the currently druggable genes in the Drug–Gene Interaction database (DGIdb)[17], showing the MS data sets captured many currently druggable and other kinases in each of the cancer cohort (Fig. 1b).

**Upregulated phosphosignaling pathways**. Tumors within a cancer type show distinct gene expression subtypes[18,19], but how they differ in kinase activities and phosphosignaling remains less characterized. We examined phosphoproteomic/proteomic upregulation of ten oncogenic signaling pathways curated by The Cancer Genome Atlas (TCGA) PanCanAtlas, including the Cell Cycle, HIPPO signaling, MYC signaling, NOTCH signaling, oxidative stress response/NRF2, phosphatidylinositol 3-kinase (PI3K) signaling, transforming growth factor-β (TGFβ) signaling, receptor tyrosine kinase (RTK)/RAS/mitogen-activated protein kinase signaling, TP53, and β-catenin/WNT signaling pathways[1]. For the six cancer types with phosphoproteomic data (Fig. 1c), pathway upregulation was calculated by statistically comparing relative pathway phosphoprotein levels against other phosphoproteins within the same sample ("Methods"). For the four other cancer types, pathway upregulation was measured by applying the same method to the relative protein level, given previous reports of high protein–phosphoprotein concordance[15,20]. We then calculated the fractions of tumors exhibiting pathway upregulation ("Methods").

Tumor samples showed clusters of different pathway upregulation within cancer types (Fig. 2a, b). BRCA tumors were separated into two predominant signaling subtypes, with one showing higher NOTCH pathway upregulation. UCEC tumors were also divided into two distinct clusters, where one showed higher TP53 upregulation. MB tumors are heterogeneous, where the RTK RAS pathway was the most frequently upregulated pathway (in 33% of the cases), followed by Cell Cycle (18%), MYC (18%), WNT (16%), and PI3K (11%). In HCC, 63% were high in HIPPO, followed by 54% in PI3K, 32% in NOTCH, and 26% in RTK RAS pathways. In PRAD, 68% of the cases showed high PI3K, followed by Cell Cycle (47%) and HIPPO (24%) pathways. In STAD, 21% showed high PI3K, followed by 19% in RTK RAS pathway. Inter-tumor heterogeneity in transcriptome-based subtypes has highlighted different oncogenic mechanisms and clinical prognosis within cancer types. Our discovery of tumor clusters of distinct phosphosignaling profiles suggests biological investigation and personalized treatment design may also need to account for their diverse pathway dysregulation.

At a single protein level, we determined differentially phosphorylated kinases between tumor and normal samples for the six cancer types with available phosphoproteomic data (i.e., BRCA, CRC, CCRCC, LUAD, OV, and UCEC) using a multivariate model adjusted for batch and clinical covariates ("Methods" and Supplementary Data 1). We identified 626 differentially phosphorylated kinase-cancer pairs (false discovery rate (FDR) < 0.05). Among these, 114 showed over 2-fold upregulation, including 47 in CCRCC, 3 in CRC, 36 in LUAD, 5 in OV, and 23 in UCEC (Fig. 2c). Top upregulated kinases in tumors include PRKRA and PRKDC in CCRCC, TLK1 and STK39 in CRC, MAPK15 and MAP2K4 in LUAD, PTK2B and NADK in OV, and PIK3C2A and MAPK15 in UCEC (Fig. 2c). Collectively, these results suggest that different cancer types differentially regulate phosphosignaling and kinase protein expression.

**Protein overexpression of currently druggable kinases**. Aberrantly overexpressed protein kinases such as HER2, epidermal growth factor receptor (EGFR), and RAS represent exploitable treatment opportunities across multiple cancer types[21]. To identify overexpressed kinases from global proteomics data, we developed the OverexPressed Protein and Transcript target

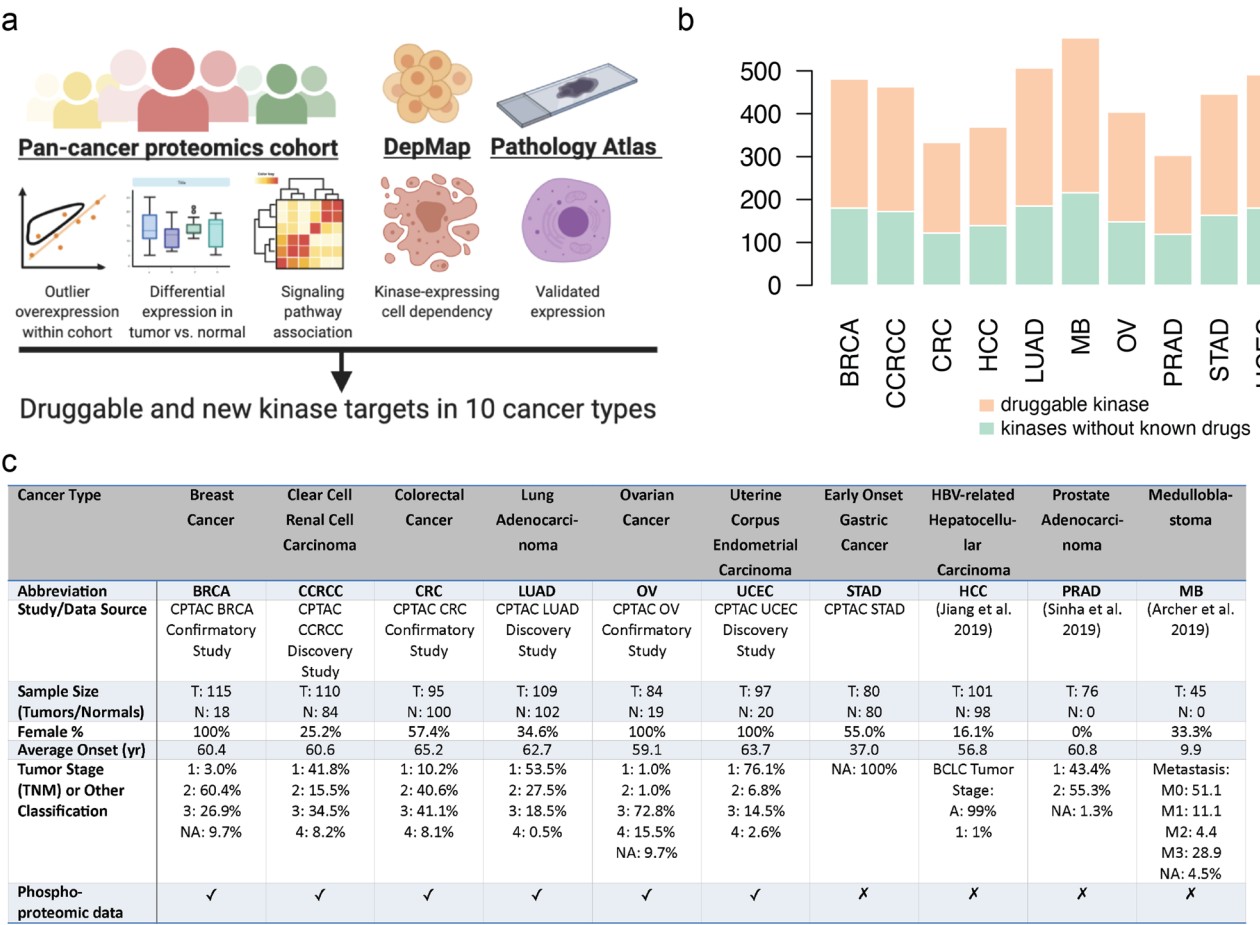

**Fig. 1 Overview of the study and the pan-cancer proteomic cohort. a** Study overview for identifying kinase therapeutic targets, combining analyses of aberrant proteins in MS proteomic data of primary tumors, cancer-cell dependencies, and immunohistochemistry evidence. **b** Distribution of quantified kinases across the global MS proteomic cohorts. **c** Summary of the sample characteristics for the ten MS global proteomic cohorts totaling 1071 cases.

Identifier (OPPTI) algorithm ("Methods" and Fig. 3a). OPPTI is designed to detect overexpressed proteins within global MS proteomic cohorts that may show varied quantitative distributions arising from different technical platforms. Existing methods to identify overexpressed markers or outliers utilize either the marker's univariate cohort distribution (z-score/interquartile range, etc.; e.g., see refs. [5,15,22]) or deviating expression from one co-expressed protein (e.g., see ref. [23]). In contrast, OPPTI computes and tests the deviation between the observed kinase protein expression level ($P_{observed}$) to that of an inferred (background) protein expression level ($P_{inferred}$) based on a $k$-nearest co-expressed protein neighbors within the proteomic data cohort to enhance robustness ("Methods"). As the single-marker approaches rely on univariate analyses, they often have to set arbitrary thresholds and would fail to identify scenarios if a high percentage of cases showed overexpression; the single-neighbor approach developed in Lapek et al.[23] overcomes this obstacle, but may be biased if the chosen neighbor marker contains noise. OPPTI's background inference is based on the commonly tested co-expression network model and the algorithm is expected to improve robustness. Moreover, compared to univariate outlier methods that often lack significance testing, OPPTI further conducts permutation testing of deviation scores with multiple-testing corrections (Benjamini-Hochberg, BH) to assess over-representation of each marker's overexpression in a cohort.

We benchmarked OPPTI performance ("Methods") by using MS ERBB2 (HER2) protein expression data from an independent global proteomics cohort of 77 primary breast tumors[15]. OPPTI

achieved 83% sensitivity and 100% selectivity ($F = 0.91$) in discerning HER2+ tumors (as determined by immunochemistry), outperforming a univariate outlier-detection method (i.e., Mertins et al.[15], Sensitivity = 66%, Selectivity = 100%, $F = 0.8$) (Supplementary Fig. 1a). In addition to testing performance at the natural rate of HER2+, we hypothesize there may be overexpressed protein markers that affect a higher fraction of tumors, i.e., an overexpressed marker present in all luminal breast tumors. Thus, we further conducted power analyses using downsized cohorts containing 40% HER2+ samples in 1000 permutations sampled with replacements. At a sample size of 30, OPPTI reached a higher average $F$ measure when using 6 ($k = 6$) co-expressed markers ($F = 0.49$), compared to using 1 ($k = 1$) co-expressed marker ($F = 0.48$, OPPTI $k = 1$, similar to ref.[23]) or compared to the univariate approach ($F = 0.40$, Mertins et al.[15]). At a cohort size of 50, OPPTI achieved an average $F = 0.56$ and 0.55 ($k = 6$ and 1, respectively), compared to that of 0.43 by the univariate approach (Supplementary Fig. 1b). When the sample size is larger than 50, OPPTI's multi-marker approach ($k = 6$) consistently outperformed the other methods. These comparisons demonstrate OPPTI's advantage in smaller and high-positive-fraction cancer proteomic cohorts compared to univariate or single-neighbor approaches.

We further benchmarked the OPPTI algorithm and other outlier-detection methods using synthetic data, by simulating log2-transformed expression values of 1000 genes for 100 samples where the benchmarked overexpressions are designed at different protruding expression levels ($\mu_{protrude}$), which determine the

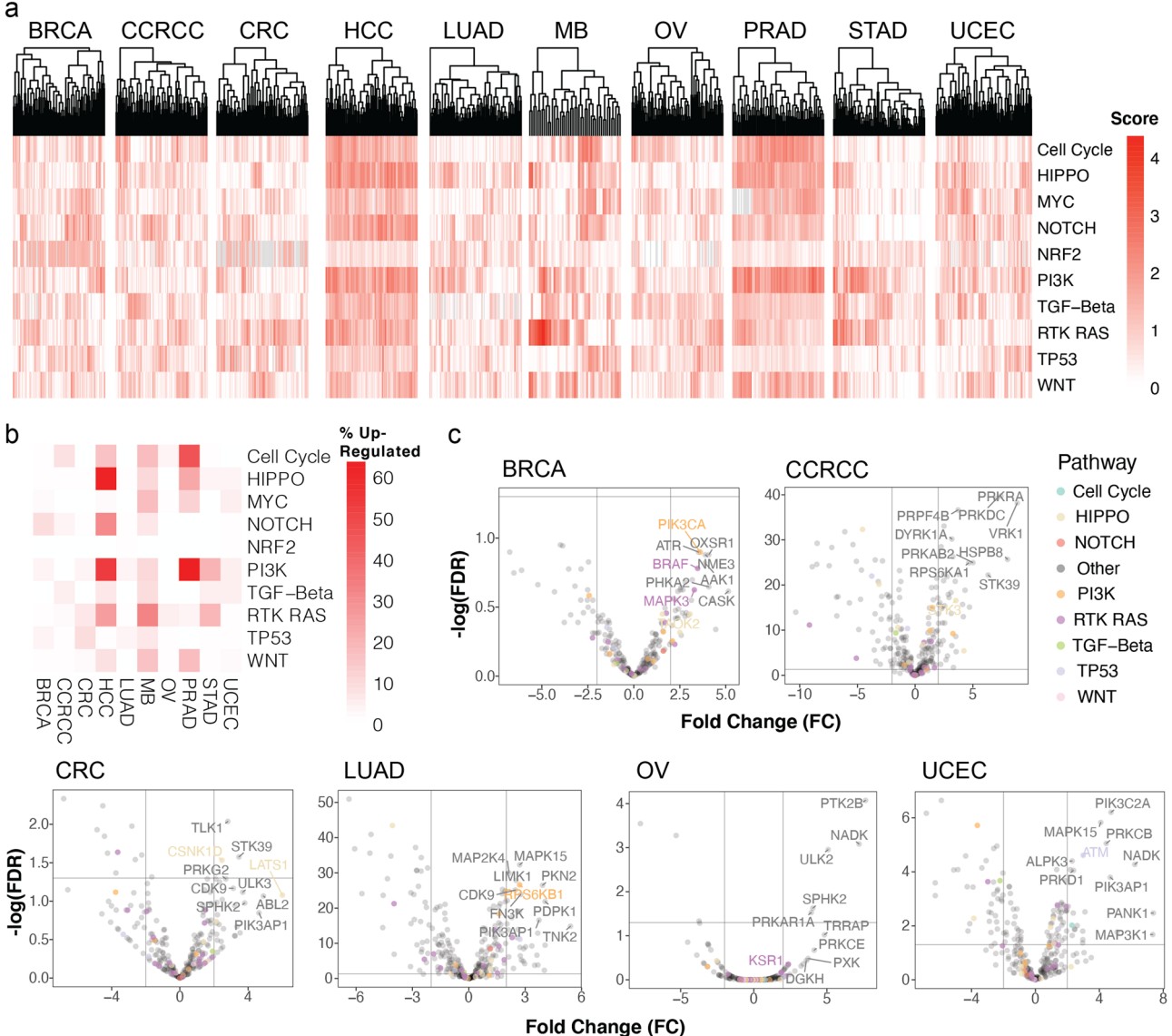

**Fig. 2 Clustering and regulation of oncogenic signaling pathways and kinase phosphoproteins. a** Hierarchical clustering of phosphoproteins in oncogenic signaling pathways across ten cancer cohorts. The color intensity indicates the pathway activity scores ("Methods"). **b** Fractions of tumors showing alteration of protein/phosphoproteins in the signaling pathway compared to other protein/phosphoproteins (p-value < 0.05). The color intensity represents the percentage in a given pathway activity. **c** Volcano plots of differentially phosphorylated kinases in the oncogenic signaling pathways. Significant markers are labeled and the chosen phosphosites representing the kinase phosphoprotein levels are in the Supplementary Data.

positive samples ("Methods"). The power analyses were conducted in downsized data containing different rates of positive samples, $N_{\text{positive}} \in \{20\%, 40\%\}$, randomly sampled with replacement through 100 permutations. In simulated data sets with $20 \sim 100$ samples and varying levels of $\mu_{\text{protrude}}$ and $N_{\text{positive}}$, the OPPTI multi-marker approach (OPPTI, $k = 4$) consistently outperformed the single-marker methods (Mertins et al.[15] and Huang et al.[15,22]) in identifying positive samples and show similar performance with the OPPTI single co-expressed marker approach (OPPTI, $k = 1$, which is conceptually similar to ref. [23]). For example, at $\mu_{\text{protrude}} = 5$ and $N_{\text{positive}} = 20\%$, and sample size = 60, the $F$ measures were 0.87 for OPPTI $k = 4$, 0.86 for OPPTI $k = 1$ (similar to Lapek et al.[23]), 0.75 for Mertins et al.[15], and 0.54 for Huang et al.[22]. Benchmark results using the synthetic data set demonstrated OPPTI's improved accuracy by leveraging co-expressed markers (Supplementary Fig. 1c). In terms of computational complexity, as expected, a univariate model (e.g., Mertins et al.[15]) consistently outperformed OPPTI. Nevertheless, when

applied to detect a single marker in a cohort of 77 samples, the algorithms showed manageable running times <30 ms (on average) in real and synthetic data sets on a laptop with a central processing unit (CPU) of 2.3 GHz (Supplementary Fig. 1d, e).

Applying OPPTI to the 10 proteomic cohorts, we identified 463 cancer-kinase pairs that showed significant enrichment of marker overexpression (FDR < 0.05), ranging from 24 in CRC to 64 in UCEC. We found 13 currently druggable kinases that showed a pan-cancer pattern of protein overexpression, i.e., affecting over 10% of cases in at least five cancer cohorts (Fig. 3b, c and Supplementary Fig. 2a). Although the only approved indications for ERBB2 (encoding for HER2) inhibitors are BRCA and STAD[24,25], we found a substantial fraction of other tumors—including 18% of MB, 16% of PRAD, and 15% of HCC—which overexpressed ERBB2. EGFR protein was found to be not only overexpressed in 17% of LUAD but also 18% of MB cases. We also identified cyclin-dependent kinase-6 (CDK6) showing over-expression across cancer types, primarily overexpressed in MB

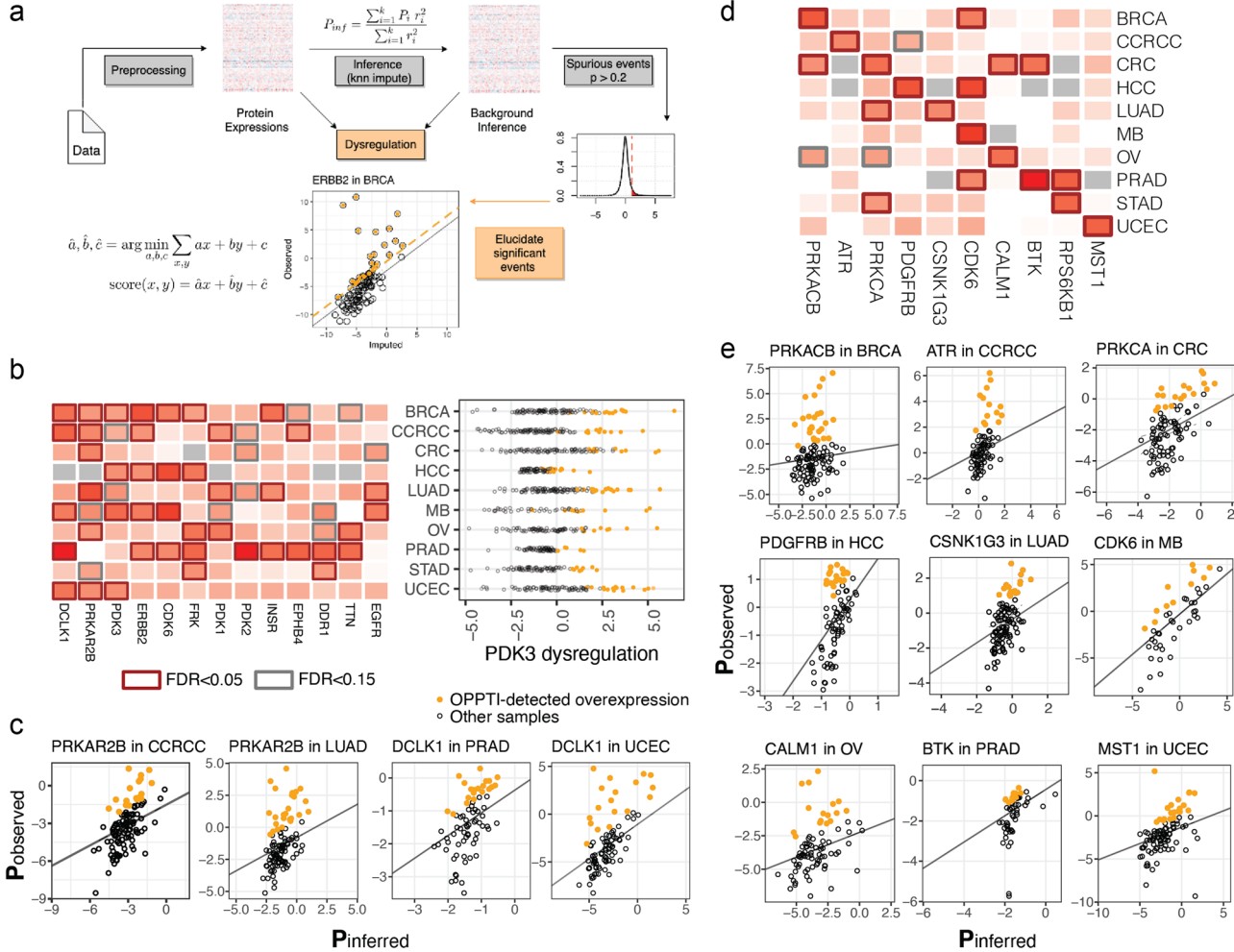

**Fig. 3 Overexpressed kinases across cancer types. a** The workflow of the OPPTI algorithm. A given marker's background expression is inferred through its nearest neighbors. The OPPTI overexpression score is measured by the deviation of observed expression from the background inference (*k*-nearest-neighbor imputation), as indicated by *score(x,y)* which calculates the distance of a point located at *(x,y)* to the best regression line fitting data via estimated $\hat{a}$, $\hat{b}$, and $\hat{c}$. For each cohort, the overexpression scores of non-dysregulated markers establish a $P < 0.05$ cutoff to define overexpressed cases. **b** Left: druggable kinases with pan-cancer overexpression, color intensity indicates the percentage of overexpressed cases. Right: PDK3 kinase's protein abundance level across cases within cancer type cohorts. Based on the OPPTI algorithm that considered a per-sample inferred expression level from co-expressed markers, the overexpression events do not completely overlap with the highest expressed samples. **c** The breakdown of DCLK1 and PRKAR2B kinase overexpression in individual cohorts, as identified by the deviation of observed expressions (*y*-axis) from the background inference (*x*-axis) and a cutoff threshold (not shown). Overexpressed cases are colored. **d** Druggable kinases showing cancer-specific patterns of overexpression. **e** Cancer-specific kinase overexpression in individual cancer cohorts.

(24%), HCC (23%), BRCA (17%), and PRAD (16%) cases (FDR ≤ 0.026). Doublecortin-like kinase 1 (DCLK1) was overexpressed in 27% of PRAD, 21% of CCRCC, 19% of both BRCA and UCEC cohorts, and 18% in MB tumors (FDR ≤ 0.021).

We further investigated the overexpressed kinases that exhibited a more cancer-specific pattern, finding 167 kinases overexpressed in over 10% of 1 cancer cohort where it showed 2-fold higher overexpression rates than other cancer types' average. The top cancer-specific overexpressed targets included Bruton's tyrosine kinase (BTK) in PRAD (overexpressed in 27% of PRAD tumors, FDR = 2.6E − 5), CDK6 in MB (24%, FDR = 1.5E − 3), platelet-derived growth factor receptor-β (PDGFRB) in HCC (22%, FDR = 7.7E − 5), and RPS6KB1 in STAD (20%, FDR = 1.7E − 4; Fig. 3d, e and Supplementary Fig. 2b). In gynecological and breast cancers, we found overexpressed PRKACB in BRCA (22%, FDR = 3.7E − 5), MST1 in UCEC (21%, FDR = 1E − 4), and CALM1 in OV (19%, FDR = 1.3E − 3). These overexpressed proteins denote molecular dysregulations common across or

specific to cancer types, and their oncogenic potential remains to be investigated.

It is possible that overexpressed kinases overlap with differentially expressed ones or they represent a new subset of targets. We performed a standard differential expression analysis (paired, tumor vs. adjacent normal control) to identify the kinases differentially expressed in tumor and compared the differentially expressed kinases to the overexpressed kinases identified by OPPTI. Only 16.6% (49 out of 295) of the significantly overexpressed kinase proteins (Protein overexpression rate > 10 and FDR < 0.05) are also significantly differentially expressed in tumor (logFC > 1 and FDR < 0.05), where many are uniquely identified by OPPTI's overexpression test (Supplementary Fig. 3a, b). We also found similar results with phosphorylation levels, where only a fraction 18.5% (17 out of 92) of the OPPTI-identified hyperphosphorylated phosphosites were also differentially phosphorylated in tumors (logFC > 1 and FDR < 0.05) (Supplementary Fig. 3c, d). Notably, many known kinase protein targets that

OPPTI identified, including ERBB2 in BRCA, CDK6 in HCC and BRCA, and EGFR in LUAD and CRC, were not found to show differential expression in tumor. OPPTI shows a unique capacity to identify proteins showing outlier expression in a fraction of cases and to reveal overexpression targets tailored to tumor subsets.

**Comparison between DNA, RNA, and protein-level alterations**. Protein-level overexpression can confirm the downstream molecular impacts arising from genomic alterations, but they may also arise posttranscriptionally[26,27] and represent new treatment opportunities. To investigate these two possibilities, we analyzed protein-level kinase overexpression patterns in conjunction with corresponding genomic alterations in the same cancer types. For the corresponding nine adult cancers in TCGA cohorts containing a total of 4188 tumors[1], we identified the fraction of cases showing prioritized somatic genomic drivers in the same kinases, including PanCanAtlas-prioritized[1] driver mutation, fusion, or copy number amplification (CNA) events. We then compared the fraction of samples showing DNA-level genomic alterations (the rate in each cohort abbreviated "DNA"), RNA-level overexpression ("RNA") to those showing protein overexpression ("PRO") detected by OPPTI (Fig. 4a and Supplementary Data 2).

We found 19 kinases with genomic alterations supported by evidence of protein overexpression in the same cancer types. As expected, PIK3CA showed the highest median genomic alteration rate (21%) across cancer types (Supplementary Fig. 4a), with varying protein overexpression rates across cancer types, including BRCA (Genomic alteration rate, DNA = 36%; PRO = 4%), OV (DNA = 24%, PRO = 6%), and UCEC (DNA = 51%, PRO = 1%), possibly reflecting the PIK3CA hotspot mutations that rarely co-occur with protein upregulation. ERBB2 genomic alteration rate was validated by substantial protein overexpression events in BRCA (DNA = 13%; PRO = 23%) and UCEC (DNA = 8%; PRO = 10%). Notably ERBB2 also showed protein upregulation in CCRCC, where it lacked genomic alterations (DNA = 0%; PRO = 15%). FGFR1 showed similar genomic amplification and protein overexpression rates in BRCA (DNA = 11%, PRO = 13%), yet in LUAD showed high protein overexpression rates (PRO = 23%) compared to little genomic alterations (DNA = 3%) (Fig. 4a, b). These results demonstrate that, even for commonly mutated kinases, protein overexpression may occur in cancer cases without genomic alterations.

We further identified a total of 25 cancer–protein pairs that were upregulated with limited genomic alteration rates, where the kinases showed substantially higher (≥3-fold) protein overexpression rate (with $P \geq 10\%$) than genomic alteration rate (Supplementary Data 3 and Fig. 4a, b). PRAD tumors harbored most kinases showing such protein overexpression patterns, notably for CDK4, ERBB2, PIK3CB, and BRAF. CDK6 and MET showed higher alteration rates in proteomic level than the genomic level consistent across at least four cancer types. These instances included CDK6 kinase in BRCA, HCC, PRAD, UCEC, and MET kinase in UCEC, CCRCC, LUAD, and STAD (Fig. 4a, b and Supplementary Fig. 4b). Other notable overexpressed proteins with limited genomic alterations included FGFR1 in LUAD and OV along with FGFR2 in OV and UCEC. Such events suggest that upregulation of oncogenic kinase proteins may arise in considerable fractions of tumors without genomic alterations.

We next compared kinase protein overexpression to their respective mRNA overexpression by applying OPPTI using the same parameters to the RNA sequencing (RNA-seq) data available for six CPTAC cancer types ("Methods"). OPPTI identified 52 kinases with mRNA overexpression supported by protein overexpression in the same CPTAC cancer cohorts (Fig. 4c). FGFR2 was the highest overexpressed kinase in both

mRNA (mRNA overexpression rate, RNA = 21%) and protein (PRO = 23%) levels within a cancer cohort (UCEC), followed by FGFR1 (RNA = 18%, PRO = 23%, in LUAD). FGFR1 was also overexpressed in BRCA (RNA = 8%, PRO = 13%) and UCEC (RNA = 10, PRO = 7%). KIT was highly expressed at the mRNA and protein levels in BRCA (RNA = 23%, PRO = 12%) and LUAD (RNA = 23%, PRO = 17%), and PDK1 was overexpressed in CCRCC (RNA = 5%, PRO = 15%) and CRC (RNA = 8%, PRO = 10%). Concurrent mRNA and protein overexpressions were commonly observed for MET, including in CCRCC (RNA = 8%, PRO = 15%), CRC (RNA = 8%, PRO = 10%), and UCEC (RNA = 11%, PRO = 19%), where MET protein overexpressions were still detected in considerable cases without mRNA overexpression. Other kinases showing overexpressions included INSR in CRC (RNA = 5%, PRO = 11%), ROS1 in LUAD (RNA = 25%, PRO = 14%), and CDK6 in UCEC (RNA = 14%, PRO = 13%).

Notably, we also identified nine cancer–protein pairs that show substantial rates of protein overexpression ($P \geq 10\%$) with limited mRNA overexpression, denoted by ≥3-fold rates of protein overexpression compared to transcriptomic overexpression (Supplementary Fig. 4c). These kinases varied by cancer types. In CCRCC, ERBB2 had limited DNA/RNA alterations (DNA = 0%, RNA = 1%) but were overexpressed in 15% of cases at the protein level, and AKT3 also showed imbalanced rates of RNA = 3% and PRO = 11%. In CRC, we identified TAOK1 (RNA = 0% and PRO = 10%), EGFR (RNA = 3% and PRO = 14%), and STK3 (RNA = 3% and PRO = 12%). In LUAD, kinases showing disproportional mRNA and protein overexpression includes STK11 (RNA = 1% and PRO = 17%), PDGFRA (RNA = 4% and PRO = 14%), and CDK4 (RNA = 4% and PRO = 14%), and in UCEC, PDGFRB showed an RNA = 1% and PRO = 11%. The discrepancies in the mRNA and protein overexpression rates support posttranscriptional regulations of these kinases in cancer and implicate protein-level analyses are required to detect their dysregulation.

We also assessed the relationship between kinase phosphorylation and mutations, which may have potential downstream phosphosignaling effects. We first analyzed the kinase hyperphosphorylation rates (as detected by OPPTI) in each CPTAC cohort with respect to corresponding genomic alterations, including missense and truncating mutations. We note that the high PIK3CA alteration (28%) observed in the BRCA cohort corresponds to substantial hyperphosphorylation at p.S312 (13%), whereas other phosphosites including MTOR p.S1261 and PIK3R2 p.S273 showed high hyperphosphorylation rates with low mutation rates on the respective kinases (Supplementary Fig. 5a, b). Using the CPTAC samples with concurrent genomic and phosphoproteomic data, we further conducted a multivariate linear regression (adjusted for age, sex, ethnicity, and MS batches) identifying protein kinase phosphosites whose expression were associated with their missense or truncating mutations, finding one significant association between DCLK1 missenses and DCLK1 p.S364 in CRC (logFC = 7.1, FDR = 0.021). PIK3CA p.S312 phosphorylation levels were suggestively associated with missense mutation status in the BRCA cases (logFC = 3.9, FDR = 0.085), implicating phosphorylation that may be correlated with oncogenic mutations.

**Cancer dependency analysis reveals new overexpressed targets**. Following our observations of multiple overexpressed kinases in off-label cancer types (i.e., where they were not the approved treatment targets), we reasoned that upregulated proteins may represent novel drug-repurposing targets. Such a hypothesis can be supported by cancer cells of corresponding lineages showing

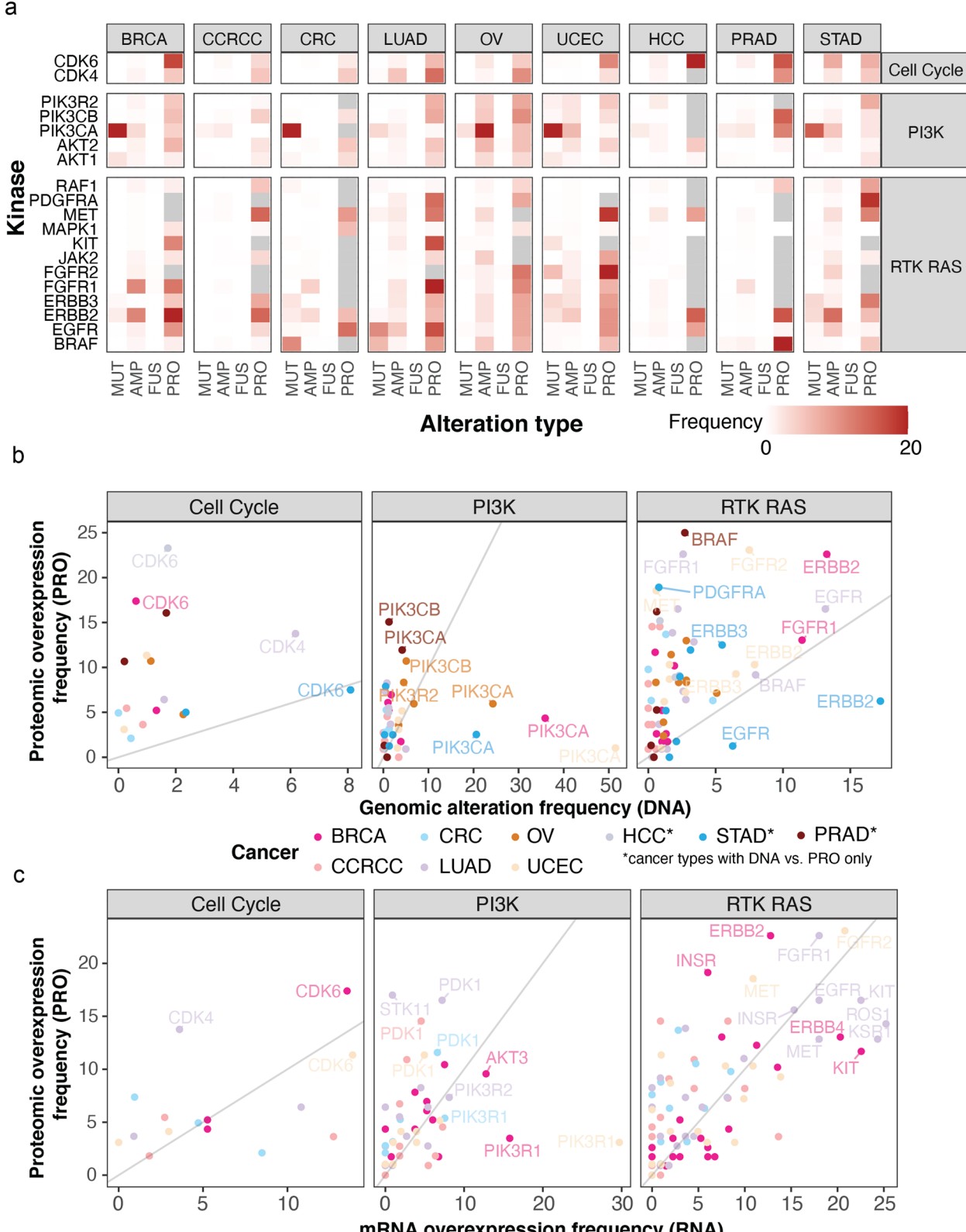

**Fig. 4 Comparison between genomic and proteomic aberrations. a** Genomic alteration (denoted by DNA) and protein overexpression (denoted by PRO) frequencies of kinases in the oncogenic signaling pathways. For each cohort, the rates of driver genomic amplification, mutation, and fusion events are individually calculated and displayed with corresponding protein overexpression rate (i.e., the percentage of overexpressed cases in the cohort). **b** Different types of genomic alteration events (amplification, mutation, and fusion) are unified (multiple alterations in the same sample are regarded as one alteration) to represent the overall genomic alteration rate within each cancer cohort (x-axis) and are compared to relative protein overexpression (y-axis). **c** mRNA (denoted by RNA) and protein overexpression frequencies of kinases in the oncogenic signaling pathways.

dependency on the overexpressed kinases. To validate OPPTI-detected overexpressed kinases, we analyzed CRISPR screen data of 625 cell lines from the Cancer Dependency Map (DepMap) project[28]. Specifically, we identified the kinases whose dependency is expression-driven, as indicated by a negative correlation between gene expression and the CERES score[29] (where a negative score indicates cancer-cell dependency) in cancer cell lines of the corresponding tissue/lineage ("Methods"). We limited the analyses to kinases with a protein overexpression rate ≥7%, i.e., affecting at least four cases in the smallest cancer cohort.

In total, we found seven overexpressed kinases with Food and Drug Administration-approved or preclinical drugs in DGIdb[17], which also showed expression-driven cancer dependency ($R \leq -0.3$, FDR < 0.05) in cancer cells of the same lineage, including one in BRCA, two in LUAD, one in MB, two in OV, and one in STAD (Supplementary Data 4). Strong expression-driven dependency was found for PDGFRA in cancer cells of the central nervous system (Pearson's correlation coefficient [$R$] = −0.53, FDR = 3.5E − 3) and stomach ($R$ = −0.8, FDR = 1 E − 3) lineages, whereas PDGFRA was overexpressed in ~18% of cases in MB and STAD. Insulin-like growth factor 1 receptor (IGF1R) was found to show expression-driven dependency by cancer cells of the ovary ($R$ = −0.6, FDR = 6.7E − 3) and lung ($R$ = −0.35, FDR = 0.04); in both cancer cohorts, we also found high IGF1R overexpression rates and hyperphosphorylation of IGF1R p.T1366 (Fig. 5a). ERBB2 showed the highest expression-driven dependency in the on-label breast cancer cells ($R$ = −0.68, FDR = 2.5E − 3) along with significant expression-driven dependency in ovarian cancer cell lines ($R$ = −0.53, FDR = 0.038; Fig. 5c). ERBB2 protein expression in OV patients was also validated by immunohistochemistry (IHC) stains of ovarian tumors from the Human Pathology Atlas[30,31] (Fig. 5d).

In parallel, we found two kinase protein targets without corresponding drugs in DGIdb simultaneously showing overexpression in ≥7% of primary patient tumors and expression-driven dependency ($R \leq -0.3$, FDR < 0.05) (Supplementary Data 5). PRKRA showed the most significant expression-driven dependency ($R$ = −0.46, FDR = 1E − 3) in lung cancer cell lines (Fig. 5a, b) and was overexpressed in 7% of LUAD cases, whereas its phosphosite PRKRA p.S130 is over-phosphorylated in 15% of LUAD. The MS-detected overexpression of PRKRA was validated by IHC stains in lung tumors from the Human Pathology Atlas[30,31] (Fig. 5d). PGK2 showed a significant expression-driven dependency ($R$ = 0.65, FDR = 0.014) in breast cancer cell lines and was overexpressed in 7% of BRCA cases. The kinase candidates identified by overlapping protein overexpression in patient cohorts and DepMap cell line vulnerability analyses present inhibition therapeutic opportunities warranting further investigations.

**Kinase overexpression associated with pathway upregulation.** Although overexpressed kinases and phosphosites represent potential activating events, observations of pathway hyperphosphorylation provide additional confidence of activated phosphosignaling and druggability. We conducted correlation analyses between the overexpressed kinase proteins and their corresponding pathway phosphoprotein (or when not available, proteins) upregulation within each cancer cohort ("Methods"). Across the 10 cancer types, we identified 21 kinases whose overexpression showed significant (Pearson's correlation, FDR < 0.05) positive correlation with its corresponding pathway's upregulation (Fig. 6a and Supplementary Data 6). The TP53 pathway was correlated with ATM overexpression in PRAD ($R$ = 0.80, FDR = 7.8e − 15) and CHEK2 overexpression in STAD ($R$ = 0.57, FDR = 7.4e − 6). The Cell Cycle pathway was

correlated with CDK2 and CDK6 overexpression in HCC ($R$ = 0.51, FDR = 2e − 5 and $R$ = 0.39, FDR = 0.01, respectively), also with CDK6 in PRAD ($R$ = 0.4, FDR = 0.025). The PI3K pathway levels were correlated with different kinases across cancer types, including AKT3 and PIK3CA in MB ($R$ = 0.47, FDR = 0.02 and $R$ = 0.55, FDR = 2.5e − 3, respectively), AKT3 and RPS6KB1 in PRAD ($R$ = 0.35, FDR = 0.025 and $R$ = 0.34, FDR = 0.04, respectively), and AKT1 and PIK3CA in STAD ($R$ = 0.36, FDR = 0.02 and $R$ = 0.33, FDR = 0.03, respectively).

For the six cancer types with available phosphosite data, we further identified 27 kinase phosphosites whose overexpression showed significant (FDR < 0.05) positive correlation with its pathway's upregulation (Fig. 6b and Supplementary Data 7). In BRCA, PI3K pathway was correlated with the hyperphosphorylation of PIK3CA p.S312 and PIK3R2 p.S273 ($R$ = 0.41, FDR = 8.2e − 4 and $R$ = 0.38, FDR = 1.1e − 3, respectively), and RTK RAS pathway was correlated with ABL1 p.S702 and BRAF p.S750 phosphorylation ($R$ = 0.35, 3.6e − 3 and $R$ = 0.3, FDR = 0.017, respectively). Among the CCRCC cases, we observed a significant correlation between the TGFβ pathway and TGFBR2 p.S573 hyperphosphorylation ($R$ = 0.55, FDR = 1.3e − 6). In CRC, the HIPPO pathway was correlated with hyperphosphorylation of its casein kinases at phosphosites CSNK1D p.T387 ($R$ = 0.31, FDR = 0.025) and CSNK1E p.S363 ($R$ = 0.43, FDR = 4.3e − 4). The TGFβ pathway was significantly correlated with TGFBR2 p.S578 hyperphosphorylation ($R$ = 0.69, FDR = 1.3e − 10) in the OV cases and with TGFBR2 p.S587 hyperphosphorylation ($R$ = 0.61, FDR = 1.7e − 7) in the UCEC cases. The UCEC cohort also showed correlation between TP53 pathway and ATM p.S1981 hyperphosphorylation ($R$ = 0.57, FDR = 3.2e − 6). Overexpressed kinases and phosphosites correlating with their pathway levels may serve as key nodes of phosphosignaling activations and provide potential biomarkers for pathway signaling activities.

## Discussion

Herein, we conducted a pan-cancer analysis of kinase and phosphosignaling vulnerability using MS proteomic and phosphoproteomic data of over 1000 cancer cases (Fig. 1), representing one of the largest cancer global proteomic studies to date. Our results showed that tumors within one cancer type showed different activation profiles (Fig. 2), which could be linked to upregulated kinase proteins or phosphosites within the signaling pathways. We identified significantly overexpressed protein kinases shared across and specific within cancer types (Fig. 3), many of which arose without concurrent genomic alterations (Fig. 4) and correlated with activated pathways (Fig. 6). Detection of such events was achieved by our newly developed algorithm, OPPTI, which demonstrated advantages compared to existing outlier methods that either utilizes only the marker's statistics within cohorts[5,15,22] or one co-expressed protein[23]. We further showed lineage-matched cancer cells showed expression-driven dependencies (Fig. 5), implicating them as cancer vulnerabilities.

Observations of overexpressed proteins provided critical validation to genomic drivers (Fig. 4). CNAs of kinases, including *ERBB2*, *FGFR1*, and *EGFR*, were correlated with overexpressed proteins in substantial fractions of the same cancer types. Meanwhile, we also identified cancer types where the same proteins were infrequently amplified, showing fusions, or mRNA overexpression—yet showing a high fraction of protein overexpression. Kinases that were rarely mutated, including MET, PIK3CB, and PDGFRA/PDGFRB, were found to be substantially overexpressed at the protein level in the same cancer types. CDK4/6 overexpression was found across multiple cancer types (Supplementary Fig. 4b), many of which lacked *CDK4/6* DNA/mRNA alterations. Such findings corroborated with preclinical

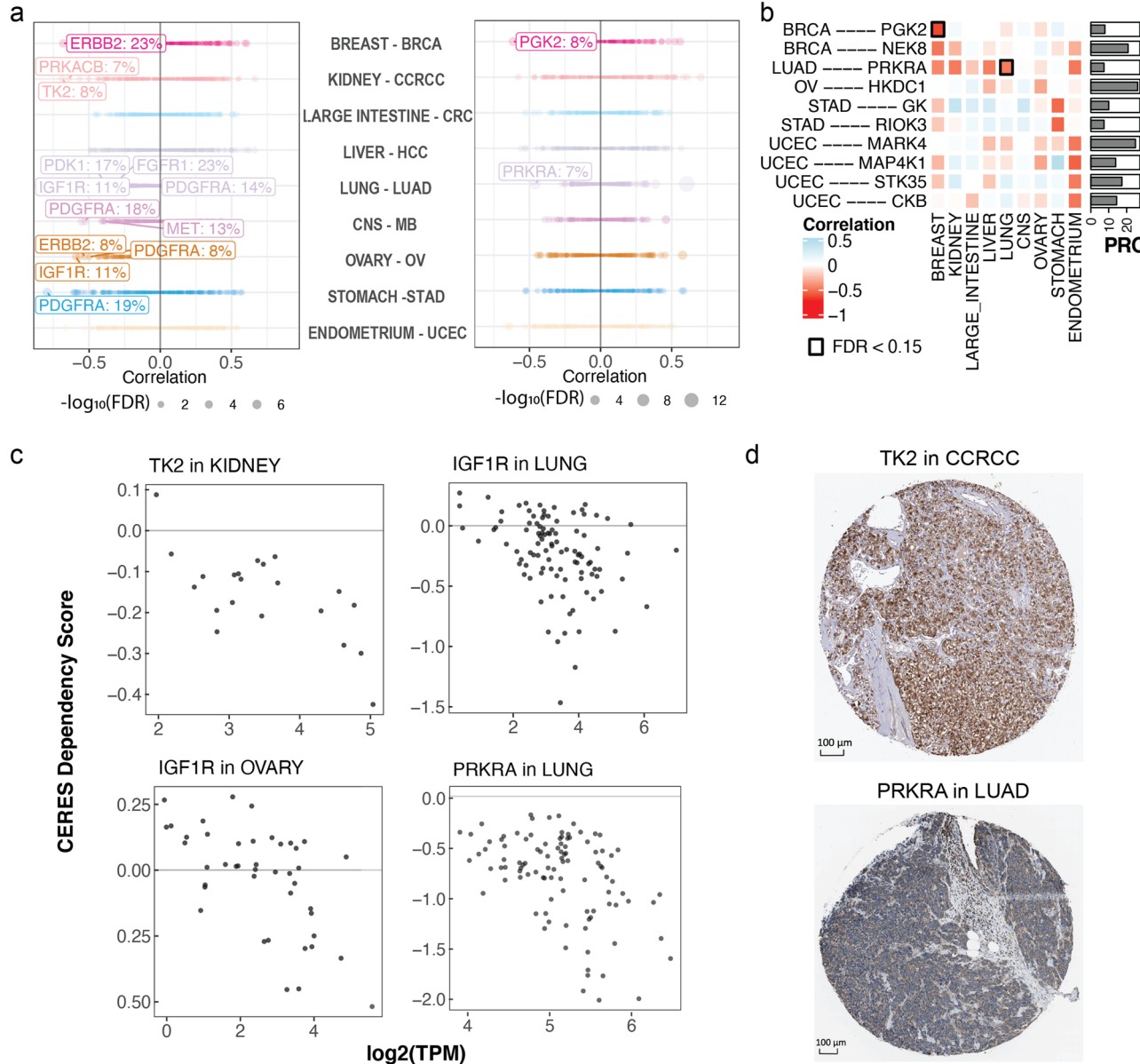

**Fig. 5 Overexpressed kinases showing expression-driven dependency. a** DepMap analyses of patient-overexpressed kinases in cancer cell lines. Left: the expression-driven dependency (correlation coefficient) of overexpressed, currently druggable kinases. Right: the expression-driven dependency of overexpressed novel kinase targets with no targeting compounds. **b** A heatmap showing the novel expression-driven dependency and the percentage of overexpressing cases in the primary tumor cohort. The side bar (PRO) displays the protein overexpression rate. **c** The correlation computed by gene expression and the cancer-cell dependency CERES scores obtained from the DepMap CRISPR screen. Each point represents a different cell line. DepMap correlations of the most significant currently druggable and novel kinases from **a** are displayed. **d** IHC staining from the Human Pathology Atlas supporting the expression of the selected druggable and novel kinase targets found in MS proteomic data to be overexpressed in samples of the corresponding cancer type. The images are obtained from The Human Protein Atlas and the staining were rated as "High" for TK2 and "Medium" for PRKRA.

and clinical evidence showing blockage of CDK4/6 inhibits the proliferation in a wide range of tumor cells[32–36]. Together, these aberrations highlight the potential of using protein expression to identify additional treatment options not readily found by genomics.

By combining evidence of patient–cohort protein overexpression and expression-driven dependencies in DepMap, we identified known and new kinase protein targets. These results demonstrated the utility of coupling patient–cohort findings and in vitro perturbation data to facilitate target discovery. The expression-driven dependency analyses had caveats. Some findings of kinase dependency derived from cell lines may not generalize to primary tumors affected by non-cell-autonomous

factors in the tumor microenvironments (e.g., stromal and immune factors). Although our findings afford promising candidates, validation using larger cohorts is required to establish protein biomarkers that can predict treatment response. We note that several identified protein targets were independently verified using IHC data of tumors from the Human Protein Atlas[30,31].

This study is one of the first pan-cancer proteomic studies[37] leveraging recently available global MS proteomic cohorts. Given the diverse sources of data sets, we utilized a standardized quality-control and normalization procedure accounting for characteristics resulting from different proteomic workflow and quantification techniques. Subsequently, we focused on reporting summary statistics and promising targets identified by OPPTI,

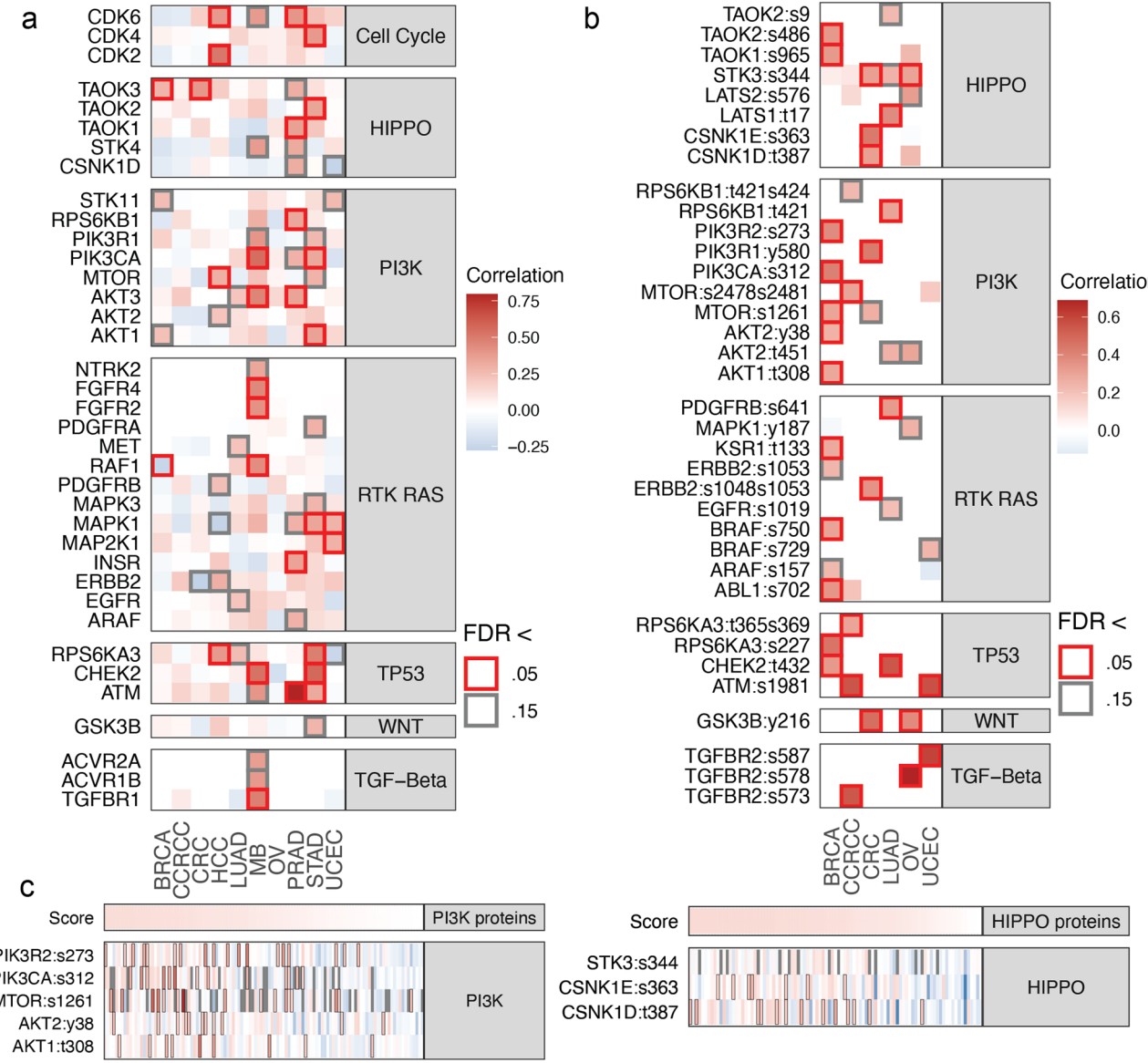

**Fig. 6 Overexpressed kinases correlated with phosphosignaling pathways. a** Overexpressed kinases that correlate with its associated pathway level in the cancer cohort. The heatmap includes all kinases that exhibit a suggestive (FDR < 0.15) and positive (R > 0) pathway correlation in at least one cancer type. **b** Oncosignaling pathway kinases whose phosphorylation correlates with pathway activity across the cases. The heatmap includes all kinase phosphosites that exhibit a significant (FDR < 0.05) or suggestive (FDR < 0.15) and positive (R > 0) pathway correlation in at least one cancer type. **c** Phosphosites significantly associated with the activities of the PI3K pathway in BRCA and the HIPPO pathway in CRC. Cases are ordered by the pathway scores.

which can adjust to characteristics within each cancer cohort. However, direct cross-cancer type comparisons will require further harmonization. Coordinated efforts, such as those undertaken by CPTAC to ensure reproducibility of results across laboratories[38] and uses of common reference samples, will be essential to enable direct proteomic comparison of one cancer type to other cancer types. Further, although the baseline proteomes and phosphoproteomes highlight potential druggable opportunities as treatment hypotheses, the protein and phosphosignaling effects of targeted treatment remain to be validated. Comparative genomic studies revealed, e.g., clonal evolution in treated tumors, where cells carrying resistance mutations expand through treatment and result in recurrence[39,40]. Proteomic/phosphoproteomic comparison of pre- and posttreatment patient tumor samples or treated vs. untreated patient-derived models is required to reveal the systematic changes underlying treatment response and resistance[41,42].

In summary, we provide a landscape of overexpressed druggable and novel kinase targets across ten cancer types. Genome-based medicine provides tailored treatment plans based on an individual's unique genetic alterations[43]. Our study demonstrates the power of proteomic analyses in revealing new treatment targets and the need to incorporate protein-level information into the precision medicine paradigm.

## Methods

**Data sources, download, and standardized normalization**. Data used in this publication were generated by multiple efforts, including the Children's Brain Tumor Tissue Consortium, part of the Gabriella Miller Kids First Pediatric Research Program, and the CPTAC (NCI/NIH). Proteomic data sets of Breast Cancer Confirmatory Study, Ovarian Cancer Confirmatory Study, Uterine Corpus Endometrial Carcinoma Discovery Study, Colon Cancer Prospective Study, CCRCC Discovery Study, Early-Onset Gastric Cancer Study, Lung Adenocarcinoma Discovery Study, and the Pediatric Brain Cancer Pilot Study were downloaded from the NCI CPTAC. Data for 45 MB cases were downloaded from Archer et al.[13]. In addition, proteomic data sets of HCC were downloaded from the PRIDE

database (www.ebi.ac.uk/pride/archive, accession numbers PXD006512 and PXD008373)[44]. Finally, the PRAD data set is downloaded from UCSD's MASSive database under the accession MassIVE: MSV000081552 at ftp://massive.ucsd.edu/MSV000081552. The overview of the proteomics/phosphosite data sets, including the number of tumor/normal samples, quantified proteins, phosphosites, and kinase proteins, is given in Supplementary Data 8.

We examined the data distribution of each cancer proteomic cohort and performed a standardized normalization procedure for each data set. Each sample within a cancer cohort is normalized by its median absolute deviation (MAD), i.e., MAD is set to 1, so that every sample across the data sets are normalized to unit MAD. We also filtered out protein markers with high fractions (at least 20%) of missing values.

The RNA-seq data sets available for six CPTAC cohorts are downloaded from https://portal.gdc.cancer.gov/. For each cancer cohort, we used the log2 normalization on the FPKM (fragments per kilobase of exon per million mapped fragments)-normalized RNA-seq counts and filtered out genes showing no expression in at least 90% of the samples. The overview of the mRNA data sets, including the numbers of tumor/normal samples, quantified genes (including noncoding RNAs), and kinase encoding genes, is given in Supplementary Data 9.

**Tumor oncogenic signaling pathways and differential activation.** For each cancer type, we analyzed the upregulation of oncogenic signaling pathways by statistical tests. To determine the activation of a certain pathway in a given sample, we conducted a one-sided Kolmogorov–Smirnov (KS) test between the sample's phosphoprotein/protein expressions from the pathway proteins and those from other proteins detected in the sample. Then we computed the pathway activity score from the $p$-values, i.e., $Score = -\log(p)$. The numbers of samples that have a significant score (Score $\geq -\log(0.05)$) were then used to calculate the fraction of the cohort showing aberrant pathway phosphoprotein levels.

**Differentially expressed phosphoproteins/phosphosites.** We tested the differential abundance level of the phosphosite showing the highest connectivity in a protein (as determined by the ConnectivityBasedCollapsing function WGCNA). For each cancer cohort, we performed a paired (tumor-normal) analysis to identify differentially expressed phosphosites by using the *limma* R package (v3.40.6). We corrected our paired differential analyses for confounding variables arising from batch effects (tandem mass tag (TMT) batch, sequencing center/operator/date) or from demographics (age, gender, ethnicity, and race) and the resulting $p$-values were multi-testing corrected using the BH procedure for FDR.

**Detection of overexpressed dysregulated kinase.** To identify overexpressed kinases based on high-throughput quantitative proteomics data, we developed the OPPTI method. OPPTI first computes an inferred protein expression level $P_{inferred}$ in each tumor sample based on a weighted $k$-nearest-neighbor algorithm, where the nearest features are the abundance level of other co-expressed proteins, after removing the outlying expression values. That is, for each protein we removed the expression levels that were at least 1.5 interquartile ranges higher than the third quartile, in order to prevent inference bias. We note that we removed these highly expressed outliers only for calculating the background expression levels; for all other analyses in our study, we used the full expression data unless otherwise indicated. Then, for each tumor sample $i$ and protein $j$, OPPTI calculates an overexpression score $O_{ij}$ to represent the extent of kinase dysregulation based on its deviation of the observed value $P_{observed}$ from the inferred one. More precisely,

$$O_{ij} = f(P_{ij,observed}, P_{ij,inferred}) = f\left(P_{ij,observed}, \sum_{n=1}^{k} P_{in} r_{jn}^2 \Big/ \sum_{n=1}^{k} r_{jn}^2\right), \quad (1)$$

where $r_{jn}$ is the weight coefficient (proximity) of the $n$-th nearest neighbor of the protein $j$ and $P_{in}$ is the expression of that neighbor protein in the same sample $i$; the function $f$ measures the dysregulation as the deviation between observed and inferred expression levels corrected by the cohort-wise difference. Let every tumor sample be represented as a point in the two-dimensional space (i.e., scatter plot) with coordinates in "observed" and "inferred" expressions of the protein $j$; then the dysregulation measure ($f$) of the given sample $i$, which is located at ($P_{ij,observed}$, $P_{ij,inferred}$), is its shortest distance to the regression line computed between observed vs. inferred expressions of the protein $j$ over all cases.

The distribution of the OPPTI scores from all cases in a cohort is then used to establish a threshold to distinguish significantly overexpressed proteins from spurious deviations that might happen by chance. This enables us to discern overexpressed kinases that may be putatively targeted for each sample. For this, we used the KS test and determined the proteins whose inferred expressions deviate from the observed values with low statistical significance ($p > 0.2$). The OPPTI scores of these proteins were used as a background distribution and the threshold is set as its 95th percentile ($p < 0.05$). Finally, we define an "overexpression" event if the OPPTI score exceeds that significance threshold.

We used permutation test to evaluate the statistical significance of each marker's potential enrichment of overexpression events within each cohort. For a given cancer cohort, we permuted the dysregulation scores within every sample between the kinase proteins and computed the null overexpressions from the permuted data. We iterated this process $N = 100 \times$ number of samples times (to

account for different degrees of freedom due to different cohort sizes), then used the null overexpressions from all iterations to build the permutation distribution. For each marker $x$, the $p$-value $p(x)$ of OPPTI overexpression OPPTI($x$) was then computed by the probability of observing this overexpression by chance, i.e., $p(x) = \frac{1}{N}\sum_{i=1}^{N} I\{Null(i) > OPPTI(x)\}$, where Null($i$) was the $i$-th null overexpression accumulated in the permutation distribution.

**Power and comparison benchmarking of OPPTI.** As a baseline validation, we used the HER2+ clinical status in the retrospective BRCA cohort containing 77 samples, and for each sample we calculated the dysregulation score from the ERBB2's nearest-neighbor proteins based on the aforementioned procedure. Then, we evaluated the sensitivity/selectivity performance of our method by the different overlaps between the true HER2+ samples and the outliers called at different (sliding) levels of the dysregulation score threshold. Similarly, we obtained the same performance curve for the univariate approach by discerning HER2+ samples at different levels of the ERBB2 expression. We reported the $F$ measure (the harmonic mean of precision and recall) corresponding to each method's threshold obtained at the value of the evaluated expressions/dysregulation scores of the ERBB2 protein. To benchmark our method for exploiting multiple markers, we performed the same analysis by using only one nearest neighbor, which is conceptually similar to the one co-expressed marker model proposed in Lapek et al.[23]. To further evaluate each method's performance in small cohorts, we performed balanced undersampling (with replacement) for 10, 20, 30, 40, 50, and 60 samples (100 times) from the 77 cases and reported the average $F$ measures by bootstrap aggregation.

We also used simulated data to assess the methods performance under various scenarios, by controlling the rate of positive ($N_{positive}$) samples that overexpress a synthetic biomarker and the level of protruding expressions ($\mu_{protrude}$) imposed by that biomarker. We performed balanced undersampling (with replacement) for 10, 20, 40, 60, 80, and 100 samples from the log2-transformed gene expressions independently simulated for each permutation and for each scenario ($\mu_{protrude}$ and $N_{positive}$ selection), then tested the methods consistently on the same permuted data for discerning the *biomarker+* samples and reported corresponding $F$ measures by bootstrap aggregation. To simulate protruding expressions, we added to randomly determined (*biomarker+*) samples a random Gaussian noise with mean $\mu_{protrude} \in \{1, 5\}$ and SD of 1.6.

We evaluated the computational complexity of the algorithms in terms of running times. All benchmarking analyses with real and simulated data are carried out on a hardware with Core(TM) i9 CPU @2.3 GHz, 16 GB memory, and Mac OSX 11.4.

**Expression-driven dependency using DepMap data.** We utilized the DepMap Public 21Q1 release from the DepMap Project[28], which contains the Achilles data set and results of CRISPR-knockout screens for 18,119 genes in 808 cell lines, which includes both cancer and normal cell lines. For each of the 683 kinase genes previously identified as druggable targets, gene expressions and corresponding CERES dependency scores were pulled. CERES is a computational method developed by Meyers et al.[29], which estimates gene dependency levels derived from CRISPR-Cas9 essentiality screens and factors in the possibility of an increase in false positives in copy number-amplified locations.

For each kinase gene, we stratified by tissue type the gene expressions and cancer-gene dependency scores across cell lines, and calculated the Pearson's correlation (and the corresponding $p$-value) between them. Overall, 1063 gene and cancer cell lines combinations were significant at an FDR of 0.05. We focused on gene–cell line combinations with negative correlation coefficients where the more overexpression of the kinase gene, the more that gene is needed for the cancer cell survival in knockout experiments. *ERBB2* was used as a key reference for establishing significance thresholds, as it is a known inhibition treatment target of breast cancer. *ERBB2* had a calculated correlation coefficient of $-0.68$ in breast cancer ($p = 7.2e - 6$, FDR $= 2.5e - 3$). The analysis was performed using R (v 3.6.2).

**Identification of pathway-associated kinase/phosphosites.** After we obtained the pathway's activity levels for all samples, we used OPPTI to screen every kinase protein in the pathway in terms of dysregulation and identified the related kinases/phosphosites that are dysregulated concurrently with the pathway activity across patients. By this procedure, we computed every association between all quantified kinases/phosphosites and ten signaling pathways in each of the ten cancer cohorts using a Pearson's correlation test and multiple-testing adjusted using the BH method for FDR.

**Reporting summary.** Further information on research design is available in the Nature Research Reporting Summary linked to this article.

## Data availability

Data for CPTAC cohorts can be found on CPTAC data portal: https://cptac-data-portal.georgetown.edu/cptacPublic/. In addition, the HCC cohort is available on the PRIDE database (www.ebi.ac.uk/pride/archive, accession numbers PXD006512 and

PXD008373)[44] or iProX database (www.iprox.org, accession number IPX0000937000)[45], and the PRAD cohort is deposited in UCSD's MASSive database under the accession MassIVE: MSV000081552 at ftp://massive.ucsd.edu/MSV000081552. RNA-seq data for CPTAC cohorts are available at the GDC data portal: https://portal.gdc.cancer.gov/.

## Code availability

The source code for OPPTI is available at https://github.com/Huang-lab/oppti.

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

## Acknowledgements

We acknowledge the participating patients and family who generously contributed to the data sets. We also acknowledge members of the CPTAC for helpful discussions. We thank the Center for Comparative Medicine and Surgery at Mount Sinai. This work was supported by the following grants: NCI CPTAC grant (U24CA210955), NIGMS R35GM138113, and ISMMS seed fund. The Tisch Cancer Institute and related research facilities are supported by P30 CA196521.

## Author contributions

K.H. conceived the research. A.E. and K.H. designed the analyses. A.E., S.T. and T.L. compiled and processed the pan-cancer proteomic data sets. A.E. developed the OPPTI algorithm. A.E. and S.J. conducted the bioinformatics analyses. K.H. supervised the study. A.E. and K.H. wrote the manuscript. M.D.G. and K.H. edited the manuscript. All authors read and approved the manuscript.

## Competing interests

The authors declare no competing interests.
