## [Transparent Peer Review File · Communications Biology]

Reviewers' comments:

Reviewer #1 (Remarks to the Author):

In this study, the authors identify kinase therapeutic targets from MS proteomic data, as well as phosphoproteome, of primary tumors. The study also introduces a new algorithm, OPPTI, to identify overexpressed kinases from high-throughput proteomics data. The paper is interesting and its biological findings are insightful. My comments on this manuscript are the following:

- What is the biological implication of the heterogeneity in e.g. pathway dysregulation, as plotted in 2A? The scale of the figure is also quite difficult to interpret because of the few high values, I suggest also trying log scale
- When calculating the pathway activity score, it is not clear to me why the authors need to use 2 tests (KS and Wilcoxon rank-sum test), and not only one, e.g., KS test, which is very standard. In addition, when combining the p-values via Fisher's method, I believe the resulting p-values are not independent, so summing them up is not a good strategy.
- When describing the algorithm OPTTI: in Figure 3A, the caption should contain what the notation in 3A is
- How does the OPTTI algorithm compare in performance with alternative ways of finding overexpressed proteins? The authors do mention some algorithms in references (lines 150-164), but they do not benchmark their algorithm with respect to alternatives
- The "number tables" as figures in 2B, 3B, 3D should be displayed as heatmaps
- There's no simulation study or sensitivity analysis benchmarking the algorithm; this is essential in understanding the theoretical properties of the algorithm and its behavior. The HER2+ status in BRCA study is of course informative, however a simulation study represents a more controlled environment.
- * No information is given on the performance of the algorithm

Reviewer #2 (Remarks to the Author):

Elmas and coauthors present an interesting manuscript to identify overexpressed kinases and downstream targets as putative cancer dependencies and drug targets. They also present a new computational method to analyse the data and nominate new highly-expressed kinases as putative targets. The integration of genomic alterations and mRNA and protein levels is of great interest because of the large efforts in the field to chart cancer genomics and transcriptomics that may be taken for granted to explain underlying biology. So both the existence and lack of agreement of the various data modalities as described by the authors is relevant. While interesting, there are limitations and problems in the current study regarding the data analysis and presentation. These need to be addressed in a revision.

1. The manuscript is listy. Throughout results, long lists of kinases, cancer types and pathways are presented as findings while focus on the underlying biology is somewhat lacking. A focus on fewer results and tying in the existing biological and translational knowledge for some top candidates detected by the authors would help the reader and increase confidence in the findings and would appeal to a broader audience outside the immediate group of oncogenic kinase signalling experts.

2. Many of the results are presented in very low statistical confidence, $FDR < 0.3$, that the authors refer to as suggestive and is expected to happen by chance up to ~30% of the time. The manuscript would be of higher confidence if such results were removed/deemphasized and instead the authors would focus on fewer findings with clear biological or translational rationale (with more detailed descriptions).

3. The OPPTI method is interesting but the current text does not mention if the method is new or unique in this space. Other methods should be mentioned. It also appears a little ad hoc - would a

standard statistical test of kinase and target expressions in cases vs. controls provide the same findings? The authors should consider justifying their method better. Ideally there would be some baseline comparison to convince the reader that the method is capable of novel insights beyond a naive analysis.

4. In the genomic effects acting on kinase levels, can the authors extend their analysis regarding SNVs potentially affecting kinase signalling? some of the most relevant hotspot mutations in kinases affect phosphorylation sites that may contribute to constitutive kinase activity. This may help explain the kinase activation patterns that are not observed at the transcript level.

5. Pathway analysis method is ad hoc (line 515). Why are the authors selecting two non-parametric statistical tests and then computing the average p-value of the two using Fisher's method? One test should be sufficient.

6. The OPPTI method removes highly expressed outlier proteins and genes. What is the effect of lowly expressed proteins/genes with low variance? RNA-seq studies often remove these genes that possibly represent low-level noise.

7. The IHC validation is either not fully developed or misleading. Did the authors perform validation experiments themselves or retrieve existing datasets with that information?

8. some of the statistical cutoffs are not well justified. For example for DepMap, three values are used $R < -0.2$, $FDR < 0.3$, $P < 0.012$ while ideally, just one FDR value would be preferred. Excessive use of such non-standard filtering would suggest lack of robustness.

9, minor. The methods section has a number of technical issues. Line 489: 5000 kinases are mentioned (there are ~600 human kinases). Lines 496 onwards, 56000 genes are mentioned (there are ~18000 protein-coding human genes, unless the authors also include non-coding genes?). Lines 469 onwards seem to include the same text copied over and over that may work better as a table. Other examples remain.

Reviewer 1 Comments: *In this study, the authors identify kinase therapeutic targets from MS proteomic data, as well as phosphoproteome, of primary tumors. The study also introduces a new algorithm, OPPTI, to identify overexpressed kinases from high-throughput proteomics data. The paper is interesting and its biological findings are insightful. My comments on this manuscript are the following:*

- **We thank the reviewer for the praise of our manuscript and have fully addressed the reviewer's comments below.**

1) *What is the biological implication of the heterogeneity in e.g. pathway dysregulation, as plotted in 2A? The scale of the figure is also quite difficult to interpret because of the few high values, I suggest also trying log scale.*

- **We thank the reviewer for bringing up this important point and have now added to the manuscript, “Inter-tumor heterogeneity in transcriptome-based subtypes have already highlighted different oncogenic mechanisms and clinical prognosis within cancer types, and our discovery of tumor clusters of distinct phosphosignaling profiles suggest biological investigation and personalized treatment design may also need to account for their diverse pathway dysregulation.”**
- **We also revised Figure 2A using the log scale as the reviewer suggested, which we agree better communicates the findings.**

2) *When calculating the pathway activity score, it is not clear to me why the authors need to use 2 tests (KS and Wilcoxon rank-sum test), and not only one, e.g., KS test, which is very standard. In addition, when combining the p-values via Fisher's method, I believe the resulting p-values are not independent, so summing them up is not a good strategy.*

- **We agree with the reviewer, and revised these analyses (Fig 2A-B, and Fig 6) by employing only one statistical test (KS) for calculating the pathway activity score. The change enabled a more straightforward interpretation of the result, while also resulting in the discovery of more significant outlier-associated pathways (Figure 6).**

3) *When describing the algorithm OPTTI in Figure 3A, the caption should contain what the notation in 3A is.*

- **Thanks for this suggestion. We revised the notation in Figure 3A, and accordingly improved the description in the figure caption.**

4) *How does the OPTTI algorithm compare in performance with alternative ways of finding overexpressed proteins? The authors do mention some algorithms in references (lines 150-164), but they do not benchmark their algorithm with respect to alternatives.*

- **We agree that the alternative methods were not properly demonstrated in the presented results. To incorporate this, we revised the text and added references (p. 6, line 166; p. 7, lines 171-172 and 185-190) to each method mentioned in the manuscript (i.e., *Sengupta et al.*⁵, *Mertins et al.*¹⁶, and *Huang et al.*²³), and accordingly revised the figure legends of the corresponding benchmarking results (Figure S1B-C), which are expanded based on the reviewer's suggestion. Please see the point #6 made below.**

- 5) The “number tables” as figures in 2B, 3B, 3D should be displayed as heatmaps.
- **We agree with the reviewer’s remark, and have incorporated this suggestion in the figures.**
- 6) There’s no simulation study or sensitivity analysis benchmarking the algorithm; this is essential in understanding the theoretical properties of the algorithm and its behavior. The HER2+ status in BRCA study is of course informative, however a simulation study represents a more controlled environment.
- **We thank the reviewer for raising this important point. In addition to resampling studies using HER2+ in a BRCA cohort, we simulated data sets based on different rates of positive samples that overexpress a synthetic biomarker and different levels of protruding expressions imposed by that biomarker, and then benchmarked the algorithms in attempt to investigate their behavior in different environments. The findings are added in Results and Figure S1, “We further benchmarked the OPPTI algorithm and other outlier detection methods using synthetic data, by simulating log₂-transformed expression values of 1,000 genes for 100 samples where the benchmarked overexpressions are designed at different protruding expression levels (μ_{protrude}) that determine the positive samples (Methods). The power analyses were conducted in down-sized data containing different rates of positive samples, $N_{\text{positive}} \in \{20\%, 40\%\}$, randomly sampled with replacement through 100 permutations. In simulated datasets with 20~100 samples and varied levels of μ_{protrude} and N_{positive} , the OPPTI multi-marker approach (OPPTI, k=4) consistently outperformed the single-marker methods (Mertins et al.¹⁶ and Huang et al.²³) in identifying positive samples, and show similar performance with the OPPTI single co-coexpressed marker approach (OPPTI, k=1, which is conceptually similar to Lapek et al.²⁴). For example, at $\mu_{\text{protrude}} = 5$ and $N_{\text{positive}} = 20\%$, and sample size = 60, the F measures were 0.87 for OPPTI k = 4, 0.86 for OPPTI k = 1 (similar to Lapek et al.), 0.75 for Mertins et al., and 0.54 for Huang et al. Benchmark results using the synthetic dataset demonstrated OPPTI’s improved accuracy by leveraging co-expressed markers (Figure S1C).”**
- 7) No information is given on the performance of the algorithm.
- **In addition to adding the synthetic data study, we have also added details in the HER2+ resampling study and other parts of the manuscripts for readers to assess the performance of OPPTI.**

Reviewer 2 Comments: *Elmas and coauthors present an interesting manuscript to identify overexpressed kinases and downstream targets as putative cancer dependencies and drug targets. They also present a new computational method to analyse the data and nominate new highly-expressed kinases as putative targets. The integration of genomic alterations and mRNA and protein levels is of great interest because of the large efforts in the field to chart cancer genomics and transcriptomics that may be taken for granted to explain underlying biology. So both the existence and lack of agreement of the various data modalities as described by the authors is*

relevant. While interesting, there are limitations and problems in the current study regarding the data analysis and presentation. These need to be addressed in a revision.

- **We thank the reviewer for the praise of our manuscript and have fully addressed the reviewer's comments below.**

1) *The manuscript is listy. Throughout results, long lists of kinases, cancer types and pathways are presented as findings while focus on the underlying biology is somewhat lacking. A focus on fewer results and tying in the existing biological and translational knowledge for some top candidates detected by the authors would help the reader and increase confidence in the findings and would appeal to a broader audience outside the immediate group of oncogenic kinase signalling experts.*

- **We have now abbreviated the listing of results, abbreviated description (ex. since we now use a unified FDR cutoff at 0.05, did not list significant FDR values), and edited the manuscript for a more general audience.**

2) *Many of the results are presented in very low statistical confidence, $FDR < 0.3$, that the authors refer to as suggestive and is expected to happen by chance up to ~30% of the time. The manuscript would be of higher confidence if such results were removed/deemphasized and instead the authors would focus on fewer findings with clear biological or translational rationale (with more detailed descriptions).*

- **We agree with the reviewer, and have now used the standard FDR cutoff of 0.05 in all reported overexpression results and figures from OPPTI and expression-driven dependency analyses, and accordingly redrafted the section discussing those findings.**

3) *The OPPTI method is interesting but the current text does not mention if the method is new or unique in this space. Other methods should be mentioned. It also appears a little ad hoc - would a standard statistical test of kinase and target expressions in cases vs. controls provide the same findings? The authors should consider justifying their method better. Ideally there would be some baseline comparison to convince the reader that the method is capable of novel insights beyond a naive analysis.*

- **In addition to mentioning the difference of OPPTI from existing methods that the previous version contained, we have now explicitly mentioned the studies that used an outlier approach to identify overexpression targets, "... Existing methods to identify overexpressed markers or outliers utilize either the marker's univariate cohort distribution (z-score/inter-quantile range, etc., ex. Mertins et al., Huang et al., Sengupta et al.)^{5,16,23} or deviating expression from one co-expressed protein (ex. Lapek et al.)²⁴."**

- **Further, in our HER2+/BRCA benchmarking and synthetic data benchmarking (added in response to reviewer 1) analyses, we now also name the methods to which we compare OPPTI (Fig S1). These benchmarking results are extensively described in Results, which consistently show a better performance of OPPTI by leveraging multiple co-expressed markers.**

4) *In the genomic effects acting on kinase levels, can the authors extend their analysis regarding SNVs potentially affecting kinase signalling? some of the most relevant hotspot*

mutations in kinases affect phosphorylation sites that may contribute to constitutive kinase activity. This may help explain the kinase activation patterns that are not observed at the transcript level.

- **We thank the reviewer for the suggestion and have added a new analysis and text to the Results section:**

“We also assessed the relationship between kinase phosphorylation and mutations, which may have potential downstream phosphosignaling effects. We first analyzed the kinase hyper-phosphorylation rates (as detected by OPPTI) in each CPTAC cohort with respect to corresponding genomic alterations, including missense and truncating mutations. We note that the high PIK3CA alteration (28%) observed in the BRCA cohort corresponds to substantial hyper-phosphorylation at p.S312 (13%), whereas other phosphosites including MTOR p.S1261 and PIK3R2 p.S273 showed high hyper-phosphorylation rates with low mutation rates on the respective kinases (Figure S5A-B). Using the CPTAC samples with concurrent genomic and phosphoproteomic data, we further conducted a multi-variate linear regression (adjusted for clinical variables and MS batches) identifying protein kinase phosphosites whose expression were associated with their missense or truncating mutations, finding one significant association between DCLK1 missenses and DCLK1 p.S364 in CRC ($\log_{FC} = 7.1$, $FDR = 0.021$). PIK3CA p.S312 phosphorylation levels were suggestively associated with missense mutation status in the BRCA cases ($\log_{FC} = 3.9$, $FDR = 0.085$), implicating phosphorylation that may be correlated with oncogenic mutations.”

We note that given the current CPTAC cohort sizes (i.e., ~100 per cancer type), there is limited power to analyze most single recurrent mutations with multiple testing correction.

5) *Pathway analysis method is ad hoc (line 515). Why are the authors selecting two non-parametric statistical tests and then computing the average p-value of the two using Fisher's method? One test should be sufficient.*

- **We agree and have revised these analyses (Fig 2A-B, and Fig 6) by employing only one statistical test (KS) for calculating the pathway activity score, enabling a more straightforward interpretation.**

6) *The OPPTI method removes highly expressed outlier proteins and genes. What is the effect of lowly expressed proteins/genes with low variance? RNA-seq studies often remove these genes that possibly represent low-level noise.*

- **We note that in MS proteomic data with isobaric labeling technologies, quantifications are typically measured in log ratio of (sample/reference), unlike count quantification in RNA-Seq. The resulting impact is that for lowly-expressed proteins, they are frequently filtered out for high missingness due to missing value in either sample or reference channel (as we do for proteins with $\geq 20\%$ NAs). If quantified, the reference typically is constructed by using a mixture of all samples or similar samples, such that the data follow a more normal distribution which we further applied MAD normalization.**
- **Given the use of a background distribution constructed across proteins, OPPTI seldom picks up the proteins with low variance but pinpoint the ones**

with high variance. This is demonstrated by examination of OPPTI scatter plots (ex. Fig 3E) where the range of observed protein expression is almost always wider than that of the inferred protein expression.

- **We have also now clarified that the removal of high outlier values was only done in establishing the background distribution in Methods, “We note that, we removed these highly-expressed outliers only for calculating the background expression levels; for all other analyses in our study, we used the full expression data unless otherwise indicated.”**
- 7) *The IHC validation is either not fully developed or misleading. Did the authors perform validation experiments themselves or retrieve existing datasets with that information?*
- **We used IHC stains from the tumor samples of the Human Pathology Atlas and have clarified this point whenever IHC is brought up in the main text.**
- 8) *Some of the statistical cutoffs are not well justified. For example for DepMap, three values are used $R < -0.2$, $FDR < 0.3$, $P < 0.012$ while ideally, just one FDR value would be preferred. Excessive use of such non-standard filtering would suggest lack of robustness.*
- **We agree and now use only $FDR < 0.05$ for reporting all results.**
- 9) *Minor. The methods section has a number of technical issues. Line 489: 5000 kinases are mentioned (there are ~600 human kinases). Lines 496 onwards, 56000 genes are mentioned (there are ~18000 protein-coding human genes, unless the authors also include non-coding genes?). Lines 469 onwards seem to include the same text copied over and over that may work better as a table. Other examples remain.*
- **We have now gathered the correct counts and represented the text in Table S8 and Table S9.**

Reviewers' comments:

Reviewer #1 (Remarks to the Author):

The authors have address most of the points. However, I find the benchmarking of the tool still a bit unsatisfactory. For example:

- the rationale of the "single marker approach" should be described a bit in more detail (why would be expect this to work better/worse?)
- S1B: for 23% HER2+, the performance of all tools is almost identical for small sample size, why is this? What happens for sample size larger than 80? Since 23% is the observed rate, why is 40% also relevant? From this analysis, it seems that the 3 methods plotted perform almost identical at observed frequency.
- No data on the computational performance of the tool is shown (how long does it take to run on each dataset, as well as on simulated data, as function of the simulation parameters)

Reviewer #2 (Remarks to the Author):

The authors have addressed my earlier comments and substantially strengthened their manuscript. I have no more major concerns, only a few minor recommendations are listed below.

1. some minor copy editing is perhaps due, for example line 69 : pan-cancer/cancer-specific
2. line 223, what does $P > 10$ mean? P is commonly reserved for P-value and that cannot be greater than 10.
3. similarly, protein overexpression rate is now called %P, for example on line 250 and onwards. perhaps fold-change (FC) or another acronym is more appropriate instead of P that might be confused with P-value.
4. line 365, " Kinase Overexpression Associated with Pathway Up-regulation", some of the R-values could be accompanied by P-values, assuming that all of these are $FDR < 0.05$.

Reviewer 1 Comments: *The authors have addressed most of the points. However, I find the benchmarking of the tool still a bit unsatisfactory.*

- **We thank the reviewer for providing valuable comments, and have fully addressed them below.**

1) *The rationale of the “single marker approach” should be described a bit in more detail (why would be expect this to work better/worse?)*

- **Thanks for this suggestion. We have now added to the manuscript (p. 6, lines 159-165),** “As the single marker approaches rely on univariate analyses, they often have to set arbitrary thresholds and would fail to identify scenarios if a high percentage of cases showed overexpression; the single-neighbor approach developed in Lapek et al.²⁴ overcomes this obstacle, but may be biased if the chosen neighbor marker contains noise. OPPTI’s background inference is based on the commonly-tested co-expression network model and the algorithm is expected to improve robustness.”

2) *S1B: for 23% HER2+, the performance of all tools is almost identical for small sample size, why is this? What happens for sample size larger than 80? Since 23% is the observed rate, why is 40% also relevant? From this analysis, it seems that the 3 methods plotted perform almost identical at observed frequency.*

- **Thanks for raising these points. We agree that all methods work similarly for HER2+ at 23% of the primary cohort (Figure S1B). HER2 is a marker with a profound outlier overexpression pattern at a limited fraction of samples of the 77 Breast tumor cohort, and thus do not typically require more complex methods for detection. The advantage of using OPPTI compared to univariate approach can be seen in the simulated case of a cohort with 40% HER2+ sample, or in the synthetic datasets (Figure S1B-C).**
- **We have now expanded the analyses to include the scenarios with 80, 90, and 100 samples. We have added to the manuscript (p. 7, lines 184-186),** “When the sample size is larger than 50, OPPTI’s multi-marker approach ($k = 6$) consistently outperformed the other methods.”
- **We have added to the manuscript (p. 7, lines 175-177),** “In addition to testing performance at the natural rate of HER2+, we hypothesize there may be overexpressed protein markers that affect a higher fraction of tumors, i.e., an overexpressed marker present in all luminal breast tumors.”

3) *No data on the computational performance of the tool is shown (how long does it take to run on each dataset, as well as on simulated data, as function of the simulation parameters).*

We thank the reviewer for bringing up this important point. We have now reported the running times for both real and simulated data sets, and added in the Results section (p. 8, lines 204-209) “In terms of computational complexity, as expected, a univariate model (ex. Mertins et al.) consistently outperformed OPPTI. Nevertheless, when applied to detect a single marker in a cohort of 77 samples, the algorithms showed manageable running times < 30 milliseconds (on average) in real and synthetic datasets on a laptop with a CPU of 2.3GHz (Figure

S1D-E).” and in the **Methods section (p. 22, lines 613-615)** “We evaluated the computational complexity of the algorithms in terms of running times. All benchmarking analyses with real and simulated data are carried out on a hardware with Core(TM) i9 CPU @2.3GHz, 16GB memory, and Mac OSX 11.4.”

Reviewer 2 Comments: *The authors have addressed my earlier comments and substantially strengthened their manuscript. I have no more major concerns, only a few minor recommendations are listed below.*

- **We thank the reviewer for providing valuable comments, and have fully addressed them below.**
- 1) *Some minor copy editing is perhaps due, for example line 69 : pan-cancer/cancer-specific.*
 - **We revised the text as “pan-cancer or cancer-specific” to avoid ambiguity.**
 - 2) *Line 223, what does $P > 10$ mean? P is commonly reserved for P -value and that cannot be greater than 10.*
 - **Thanks for pointing this out. We now used the abbreviation ‘PRO’ to represent the protein overexpression.**
 - 3) *Similarly, protein overexpression rate is now called %P, for example on line 250 and onwards. perhaps fold-change (FC) or another acronym is more appropriate instead of P that might be confused with P -value.*
 - **We now used the abbreviation ‘PRO’ to represent the protein overexpression and used the letter ‘p’ to represent the p-value to avoid confusion.**
 - 4) *Line 365, " Kinase Overexpression Associated with Pathway Up-regulation", some of the R -values could be accompanied by P -values, assuming that all of these are $FDR < 0.05$.*
 - **Thanks for raising this point. We have now reported the corresponding FDR values.**